# Precipitation Behavior of Carbides and Its Effect on the Microstructure and Mechanical Properties of 15CrNi3MoV Steel

**Xin Yao [1], Jie Huang [1,2], Yanxin Qiao [1,\*], Mingyue Sun [2,\*], Bing Wang [2] and Bin Xu [2]**

1    School of Materials Science and Engineering, Jiangsu University of Science and Technology, Zhenjiang 212003, China
2    Institute of Metal Research, Chinese Academy of Sciences, Shenyang 110016, China
\*    Correspondence: yxqiao@just.edu.cn (Y.Q.); mysun@imr.ac.cn (M.S.)

**Abstract:** In this study, quenching and tempering were employed to achieve the optimal match of strength and toughness of the high-strength low-alloy (HSLA) 15CrNi3MoV steel. The effect of the tempering temperature on the microstructure evolution and the carbides precipitation of the steel was also investigated using scanning electron microscopy (SEM), a X-ray diffractometer (XRD) and transmission electron microscopy (TEM). The results showed that after tempering at different temperatures, the microstructure of 15CrNi3MoV steel was tempered martensite. During the tempering process, the $M_3C$ carbides precipitated on the ferrite matrix, the needle-like carbides accumulated and grew into a short rodlike shape or a granular shape with the increase of the tempering temperature. Subsequently, the strength and hardness of the steel showed a downward trend, and the elongation and the low temperature impact toughness showed an upward trend. The tensile strength and yield strength of the steel tempered at 650 °C decreased dramatically compared with the steel tempered at 550 °C, while the elongation increased rapidly. Considering the influence of the microstructure and the carbides and the demand for mechanical properties, the optimal tempering temperature is about 600 °C.

**Keywords:** high-strength low-alloy (HSLA) steel; tempering; mechanical properties; carbides

## 1. Introduction

High-strength low-alloy (HSLA) steels are widely employed in aerospace, automobile manufacturing, building structure and ocean engineering, due to their advantages of a low cost, a high strength, easy welding and processing [1,2]. The high strength of the steel always leads to the limited impact toughness, especially at a low temperature, which restricts the application of high strength steel in some specific fields [3–8]. Some engineering parts used in a harsh environment required not only a high strength, but also high requirements on the low temperature toughness, such as some ship parts. Therefore, high strength steel with a good strength and matching toughness attracts a lot of attention [9].

In order to obtain a good match of strength and toughness, the production process, such as the smelting process [10], the optimization of the alloy system [11–13], the thermo-mechanical controlled process (TMCP) [14,15] and the heat treatment process [16,17] are often adopted. Metallurgists have carried out a lot of research to reduce the content of harmful impurity elements in steel and improved the morphology of the inclusions so as to ameliorate the strength and toughness of the HSLA steel. Electroslag casting (ESC) is a special casting process which combines the remelting, refining and casting processes. Studies [18,19] have shown that a finer solidification structure, a higher purity, a lower segregation and a better surface quality can be obtained by controlling the ESC process parameters. However, smelting is just the first process of manufacturing a product and the mechanical properties of cast steel are far worse than those of forged steel. In addition, a good strength and toughness can be obtained by adding microalloying elements to the

steel by means of solution strengthening and precipitation strengthening. By adjusting the content of the alloying elements Cr, Mo and V in the steel [20] and TMCP technology [21], the optimized microstructure can also be obtained, and the impact toughness of the high strength steel can be obviously improved. Although the addition of the alloying elements can improve the mechanical properties of steel, the production cost increases. Under the condition that the properties can be met, the most economical and suitable steel type should be selected. Heat treatment is one of the most economical and feasible methods to adjust the mechanical properties. A good toughness of high strength steel is usually achieved by adjusting different heat treatment parameters. Quenching and tempering are the most commonly used heat treatment processes. Onizawa et al. [22] studied the strengthening mechanism of carbide MX (Carbonitrides of V and Nb) and its influence on the mechanical properties of a high chromium heat-resistant steel after 873 K tempering, and found that the nano-carbide VC and NbC hardly increased after 6000 h of holding at this temperature, which significantly improved creep strength of the material. Li et al. [23] investigated the influence of the different tempering temperatures on the microstructure and the impact toughness of the G18CrMo2-6 heat-resistant steel, and revealed that in addition to the softening effect of the matrix structure, the type, morphology, size and distribution of the second phase in the tempered structure was a key factor affecting its impact properties. Du et al. [24] discovered that after tempering at 610 °C for 20 min, the $M_3C$ carbides precipitated at the grain boundaries of the Fe-Cr-Ni-Mo-V steel. Following the extension of the tempering time to 2 h, the $M_3C$ carbides gradually decomposed and a certain amount of $M_6C$ and the relatively stable MC carbides formed. These types of carbides could promote the precipitation strengthening and had little effect on the toughness and plasticity, thus a good matching of the strength and toughness could be obtained. In the quenching and tempering heat treatment, the martensite steel becomes very hard and brittle after the water-quenching [25]. So the tempering treatment is necessary to improve the toughness of the steel and obtain a better match of hardness and toughness [26]. Salemi et al. [27] focused on the mechanical properties of a NiCrMoV steel after it was tempered at temperatures in the range of 200–600 °C, they found that the yield strength (YS) and the ultimate tensile strength (UTS) decreased with the increased tempering temperature while the Charpy impact energy was improved. Zhu et al. [28] indicated that the reason for change of the mechanical properties of the steel after the tempering is because the tempering temperature increases the dislocation and the density decreases gradually, the martensite decomposes and the carbide precipitates, aggregates and grows. In the process of tempering, the carbides precipitate in the HSLA steel [4,29,30], and the different tempering temperatures and time will contribute to the differences in their types, morphology, size and distribution. Therefore, how to accurately predict the formation temperature range of the precipitation of the carbides and the evolution of the martensitic structure are vital for optimizing the mechanical properties [31].

In this study, the effect of the tempering temperatures (350–650 °C) on the microstructural evolution and the mechanical properties of the 15CrNi3MoV steel have been discussed, which can provide a theoretical basis for formulating an appropriate tempering heat treatment process for the actual large-scale castings and forgings of HSLA steel.

## 2. Experimental Procedure

### 2.1. Sample Preparation

The steel used in this study is a high-strength low-alloy 15CrNi3MoV steel, and its chemical composition is shown in Table 1. The 15CrNi3MoV steel was fabricated using an electric furnace and refined through a procedure in a ladle furnace (LF) and in a vacuum degassing furnace (VD). The ingot was maintained at 1200 °C for 6 h, and then forged into a step with the dimensions of 70 mm × 70 mm × 1100 mm. The normalizing temperature was selected to be 840 °C for 1 h, followed by air cooling. The samples were heated to 890 °C for 1 h, followed by water cooling to room temperature. According to the CCT curve of the 15CrNi3MoV steel, the temperature range of the martensite transition is 240 °C to

420 °C, and that of the austenite transition is 730 °C to 830 °C. Therefore, a wide tempering temperature range of 350 °C to 650 °C was chosen for this study. The samples were tempered at 350, 450, 550 and 650 °C for 2 h followed by air cooling.

**Table 1.** Chemical compositions of the 15CrNi3MoV steel (wt.%).

| C | Si | Mn | P | S | Ni | Cr | Mo | V | Fe |
|------|------|------|-------|-------|------|------|------|-------|------|
| 0.13 | 0.27 | 0.46 | 0.003 | 0.002 | 2.80 | 1.05 | 0.23 | 0.048 | Bal. |

### 2.2. Microstructure Analysis

The microstructure examination of the tempered and untempered samples was carried out. The samples for the optical microscopy (OM, AxioCam MRc5, Zeiss, Oberkohen, Germany) and scanning electron microscopy (SEM, quanta 600, FEI, Eindhoven, The Netherlands) analyses were polished and etched with the nitric reagent (4 mL $HNO_3$ + 96 mL ethanol). The SEM analysis was also applied to characterize the fractography of the fracture Charpy V-notch impact samples.

In order to reveal the morphology and distribution of the carbides, thin foils for the transmission electron microscopy (TEM) were prepared using a twin-jet electro-polishing method in a solution of 10 mL perchloric acid and 90 mL ethanol at 30 V and −20 °C. The microstructure analysis and the precipitate identification were performed using a FEI Tecnai Spirit T12 electron microscope (Eindhoven, The Netherlands).

The electrolytic extraction was employed to extract the carbides from the matrix in order to identify the type and composition of the carbides without interference from the matrix. The detailed procedure for the electrolytic extraction process is described in reference [32]. The X-ray diffraction (XRD) studies were carried out to measure the precipitate phase at room temperature in a PHILIPS APD-10X (Eindhoven, The Netherlands) diffractometer. The data were collected using Cu Kα radiation in the 2θ range from 30° to 90° at a scanning rate of 10°/min.

### 2.3. Mechanical Properties

The mechanical properties of the samples under the different heat treatment conditions were determined using the standard methods. The room temperature tensile and the low temperature Charpy V-notch impact samples were prepared through a tempering treatment. A testing machine (AG-100KN, Shimadzu, Kyoto, Japan) with a strain rate of 3 mm·min$^{-1}$ was employed to analyze the tensile test samples with dimensions of Φ5 mm in diameter and 60 mm in gauge length at temperatures of 25 ± 1 °C, and the elongation is measured using a mechanical extensometer. A standard pendulum-type impact testing machine (ZBC2452-C, MTS, Minneapolis, MN, USA) was used to analyze the Charpy V-notch samples with a size of 10 mm × 10 mm × 55 mm at −20 °C. For each set of the samples, the impact tests were repeated at least three times and then the results were averaged. A RB2000 Digital Rockwell Hardness Tester (Wilson, Chicago, IL, USA) was used to measure the HRC hardness of samples with a load of 150 kg and a loading time of 15 s. In addition, each effective hardness data of the sample was the average value of five points.

## 3. Results and Discussion

### 3.1. Quenched Microstructure

The OM and SEM images of the 15CrNi3MoV steel after quenching at 890 °C are shown in Figure 1. The outline of the prior austenite grains (PAGs) is faintly discernible in Figure 1a. Each PAG includes several blocks with a different bit orientation, which are composed of many slender martensite laths that are nearly parallel and closely arranged [31,33], as shown in Figure 1b. Some martensite packets are found in the PAGs with irregular shapes with a preferred orientation [34]. Generally, the lath martensite with a carbon content less than 0.2% will self-temper (a tempering phenomenon that occurs when the specimen is not completely cooled during the quenching process due to the dimensional effects) during

quenching and cooling, which leads to the precipitation of carbides [35]. However, there was no marked carbide observed at the PAG grain boundaries, the lath martensite and the interlath regions. It may be attributed to the addition of Ni (2.8 wt. %) that was favorable to form martensite and eliminate to form the carbides in the as-quenched 15CrNi3MoV steel [36].

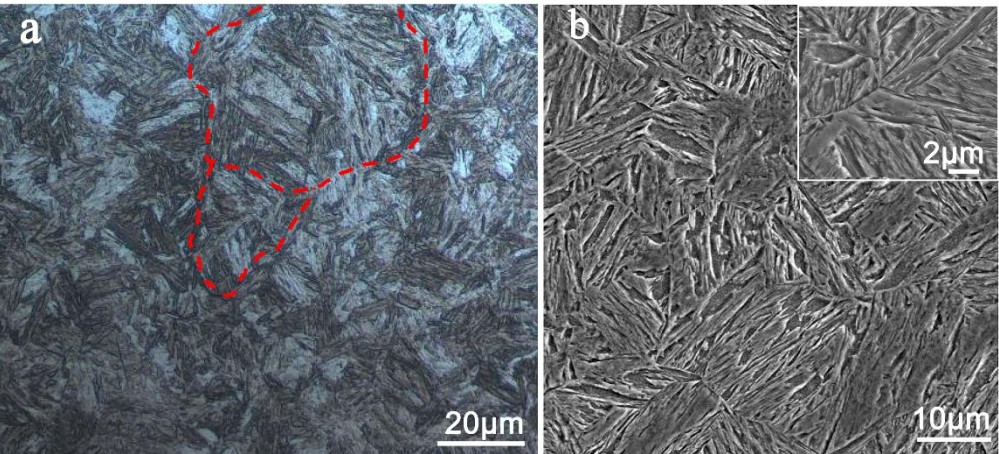

**Figure 1.** OM (**a**) and SEM (**b**) images of the quenched 15CrNi3MoV steel.

### 3.2. Microstructural Evolution during the Tempering

Figure 2 shows the SEM images of the quenched 15CrNi3MoV steel tempered at various temperatures. As shown in Figure 2a,b, the tempered martensite was obtained through tempering at temperatures of 350 °C and 450 °C. From Figure 2a, the tempered martensite exhibited a similarity to that of the quenched sample (Figure 1b), but the difference was that there precipitation of the fine carbides in the interlath regions after the tempering, as shown by the white arrows in the Figure 2. Due to the relatively low thermal activation energy of 350 °C, the carbides precipitated preferentially along the martensite lath boundaries. From Figure 2b, the martensite lath boundaries were gradually blurred, due to the enhancement of the atomic migration, the diffusion rate and the recovery effect. Meanwhile, it is evident that the carbides were gradually increased and coarsened due to the increased activity of the carbon atoms and their ability to carry out long distance diffusions.

It can be seen that the microstructure of the 15CrNi3MoV steel tempered at 550 °C was lath martensite while the microstructure of the steel tempered at 650 °C was equiaxed tempered martensite, which kept the lath martensite phase direction, as shown in Figure 2c,d. With the increase of the tempering temperature, the martensite was further decomposed into ferrite and carbides [37], and the lath grain boundary was not obvious. In addition, the recovery rate of the matrix increased significantly, which made the dislocation density decrease, the dislocation lines became straight, and the dislocation cells and intracellular dislocation lines in the matrix gradually disappeared. When the tempering temperature reached 650 °C, the matrix began to recrystallize and gradually became equiaxed crystal with a low dislocation density. Compared with the tempering at a lower temperature, the carbides that precipitated on the matrix became bigger and coarser.

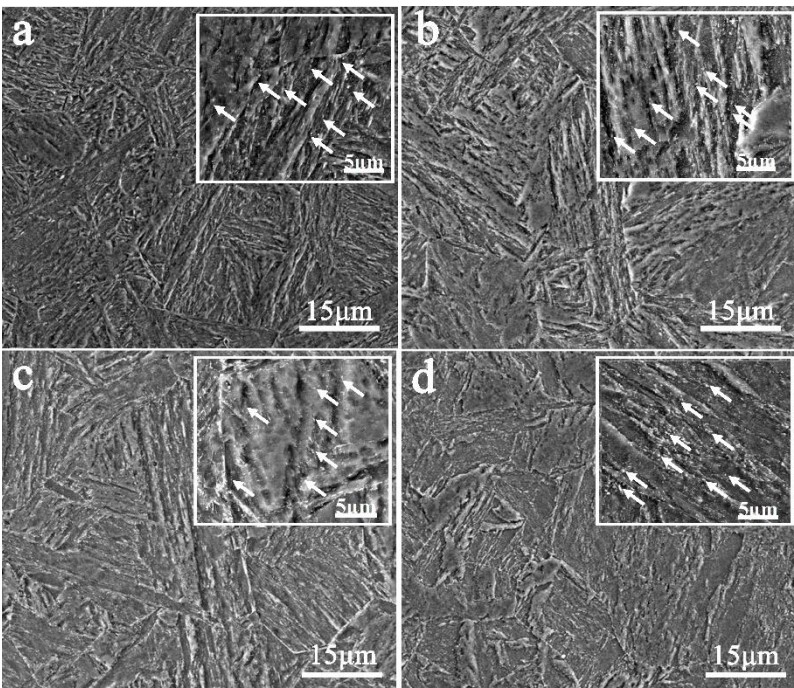

**Figure 2.** Microstructure of the quenched 15CrNi3MoV steel tempered at (**a**) 350 °C, (**b**) 450 °C, (**c**) 550 °C and (**d**) 650 °C for 2 h.

The evolution of the carbides with the increase in the tempering temperature is plotted in Figure 3. When tempered at 350 °C, the precipitates in the 15CrNi3MoV steel were acicular with a width of about 12 nm and a length about 120 nm, as shown in Figure 3a. With the tempering temperature rising to 450 °C, the carbides began to aggregate and visibly grow, and the ratio of the length to width of the carbides decreased gradually. Following the tempering at 550 °C, almost all of the carbides had changed into short rodlike or granular particles, and the small carbides disappeared gradually. When the tempering temperature reached 650 °C, the granular carbides rapidly aggregated and coarsened, and the small carbides almost disappeared. The average length or diameter of the short rodlike or granular carbides was about 70 nm. It demonstrates that the spheroidization and the growth of the carbides follow the mechanism of dissolution of the fine particles and the growth of coarse particles.

During the tempering, the martensite decomposed to form a ferrite matrix and a precipitated phase. The size, quantity and shape of the precipitates were basically determined by the tempering temperature and dwell time. In other words, thermodynamics is the key factor on the influence of the precipitation. Chen et al. [38] reported that the $M_3C$ precipitates precipitated from a bainitic ferrite matrix had a pinning effect on the dislocations when the Cr-Mo-V steel was tempered at a lower temperature, which hindered the recovery and the recrystallization of the ferrite at a lower temperature. It was reported by Sun et al. [39] that the carbides in the low carbon low alloy steel were mainly of the $M_3C$ type and preferentially located along the martensite lath boundaries. The carbides precipitated during the tempering were determined using XRD tests, the results are shown in Figure 4. The results showed that there are mainly $Fe_3C$ carbides (alloy cementite ICSD#64689) and it is found that the type of the precipitated phase is independent of the temperature. That is consistent with the claims in the existing research results [40,41]. Otherwise, we cannot deny that there may be a very small amount of VC precipitates, which also have some influence on the mechanical properties [31].

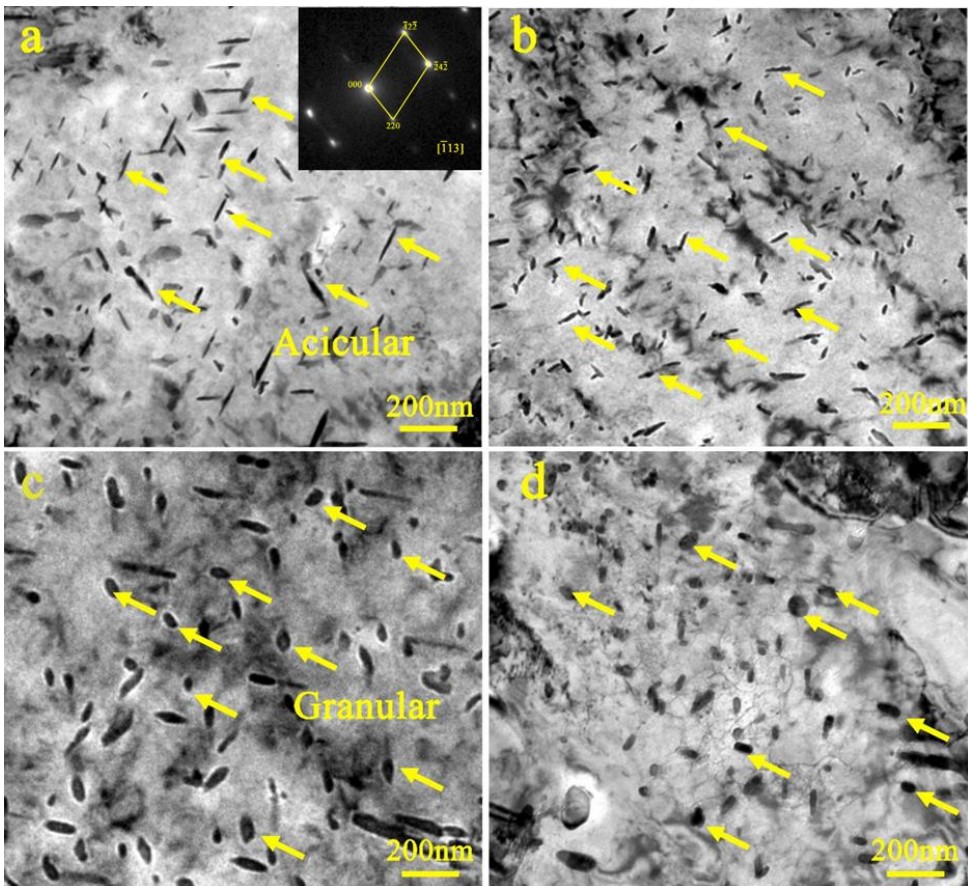

**Figure 3.** TEM diagram of the precipitates of the quenched 15CrNi3MoV steel tempered at (**a**) 350 °C, (**b**) 450 °C, (**c**) 550 °C, and (**d**) 650 °C.

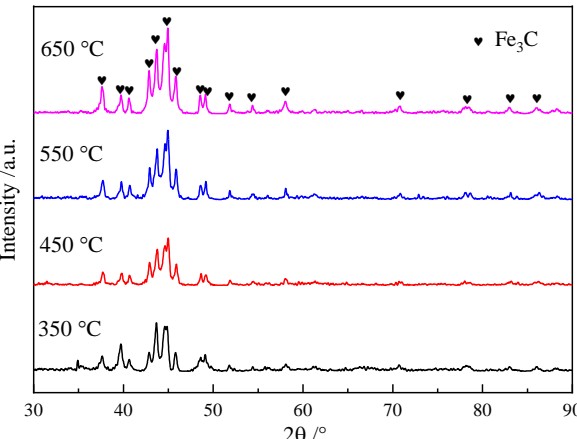

**Figure 4.** XRD pattern of the precipitate phase in the quenched 15CrNi3MoV steel at various tempering temperatures.

When the tempering temperature was lower than the experimental temperature, most carbon atoms tended to converge near the dislocation lines, and no ξ-carbides ($M_2C$, $M_4C$) precipitation occurred. That is one of the reasons why the carbides precipitated preferentially, along the martensite lath boundaries. With the increase of tempering temperature, the θ-carbides ($M_3C$) were precipitated directly from the martensite in the separated region of the carbon atoms, which are more stable than the ξ-carbides. When the newly formed carbides grew to a certain size, the coherent relationship was destroyed one after another, and the carbides were separated from the parent phase and precipitated, as shown in

Figure 3. With the increase of the tempering temperature, the diffusion rate of alloying elements increased, and the solid solubility of the carbides increased, so the size of the carbides would gradually grow and spheroidize, and the number of carbides would also decrease [24].

### 3.3. The Effect of the Microstructure on the Mechanical Properties

The mechanical properties of the quenched steel change with the variation of the tempering temperature, which is closely related to the evolution of the microstructure.

Figure 5 shows the mechanical properties of the 15CrNi3MoV steel after the tempering at different temperatures. It can be seen that the temperature had an important impact on the mechanical properties in the tempering temperature range of 350–650 °C. The yield strength (YS) and the tensile strength (TS) both monotonously decrease with the increase of the tempering temperature. Above 550 °C, the decrease in the YS and the TS of the steel are gradually pronounced. At the tempering temperature of 650 °C, the YS and TS were 752 MPa and 825 MPa, respectively. In addition, the difference between the YS and TS decreased with the increase of the tempering temperature and reached the minimum at 650 °C. This behavior was related to the precipitation of the carbide [42].

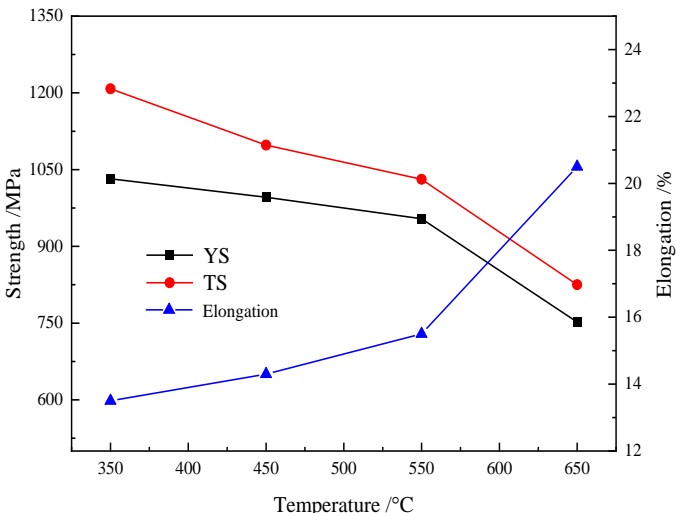

**Figure 5.** Effects of the tempering temperature on the mechanical properties of the 15CrNi3MoV steel.

The effect of the tempering temperature on the ratio of the yield strength to the tensile strength of the samples is shown in Table 2. The high yielding to the tensile ratio means that the material has a strong resistance to deformation and is not prone to plastic deformation. On the one hand, the strengthening effect of the precipitates led to the increase of the ratio. On the other hand, the dislocation density decreased with the increase of the tempering temperature, and the matrix softening effect led to the ratio reduction. At the tempering temperature below 550 °C, the strengthening effect was greater than the softening effect, which made the ratio increase; when the tempering temperature was higher than 550 °C, the strengthening effect was less than the softening effect, which made the ratio decrease. Generally speaking, the yield to tensile ratio of the low alloyed steel is between 0.65 and 0.75 at the service operating temperature [43]. The ratio of the 15CrNi3MoV steel peaked at 550 °C over 0.92, which is far more than the usual number.

**Table 2.** The ratio of the yield strength to the tensile strength of the 15CrNi3MoV steel.

| Tempering Temperature | 350 °C | 450 °C | 550 °C | 650 °C |
|:---:|:---:|:---:|:---:|:---:|
| **YS/TS** | 0.8543 | 0.9071 | 0.9253 | 0.9115 |

The response of the elongation was contrary to those of the YS and the TS. The results at temperature of 350 °C showed that the 15CrNi3MoV steel had a limited elongation of 13.5%. The elongation increased with the tempering temperature and this trend became more pronounced at the tempering temperature of 650 °C. When the temperature increased to 450, 550 and 650 °C, the dates of the elongation were 14.3%, 15.5% and 20.5%, respectively. With the increase of the tempering temperature, the martensite gradually decomposed and then the orientation of the original martensite disappeared. With the growth of the precipitated carbides, the content of the alloy elements in the martensite matrix decreased and the matrix softened, which contributed to the increase in the elongation.

Figure 6 shows the impact energy and the Rockwell hardness (HRC) of the 15CrNi3MoV steel tempered at various temperatures. The values of the impact energy of the samples tempered at 350 °C and 450 °C were 166.3 J and 168.7 J, respectively. At the tempering temperature above 550 °C, the impact energy increased dramatically, and thus, a considerable increase of energy absorbed with the increased tempering temperature was observed. However, the effect of the tempering temperature on the Rockwell hardness (HRC) was almost contrary to that of the impact energy. The hardness of the 15CrNi3MoV steel tempered at 350 °C was 39.2 HRC, and a substantial softening effect was observed for the tempered alloy. Increasing the tempering temperature to 650 °C, led to a significant decrease in the hardness to 24.2 HRC. This was due to the continuous precipitation and coarsening of the precipitates.

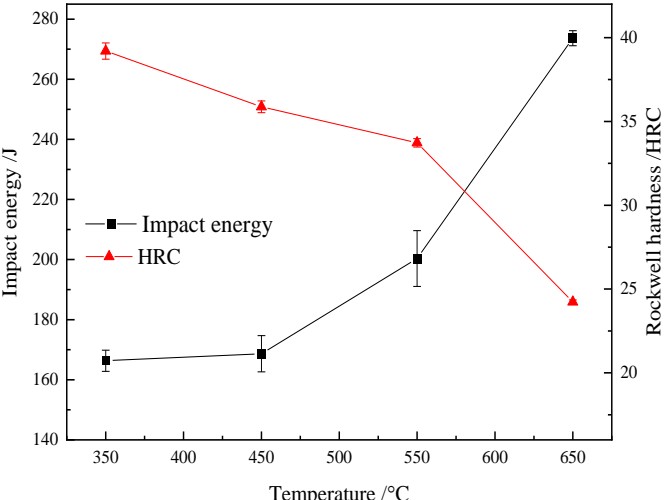

**Figure 6.** Effects of the tempering temperature on the impact toughness and the Rockwell hardness of the 15CrNi3MoV steel.

The impact energy, the tensile strength and the yield strength decreased slowly in 350–550 °C and sharply in 550–650 °C, while the opposite was observed in the elongation and the hardness. Considering its mechanical properties, the yielding to tensile ratio and the actual service conditions, 600 °C is the most appropriate tempering temperature, which can obtain a better strength and toughness at the same time.

### 3.4. The Analysis of Fracture Morphology

Figure 7 shows the morphologies of the impact fracture of the tempered 15CrNi3MoV steel at various tempering temperatures. Obviously, all of the samples after the heat treatment processes exhibit a vast number of dimples, implying a fracture mode based on the aggregation of the microvoids. As shown in Figure 7a, the fracture surface is characterized by a mixture of a cleavage fracture and a dimple fracture. This cleavage fracture is due to the precipitation of the continuous or discontinuous carbides along the interlath regions (Figure 2a). The ductile and dimple-like features were clearly observed on the fracture surface of the original material, and the diameter and depth of the dimples

with a non-uniform size were large, which was consistent with the excellent ductility of the original material in this condition. At the tempering temperature of 450 °C, the distribution of the dimples was not uniform and the volume of the dimples had no obvious change. With the increase of the tempering temperature, the number and size of dimples increased gradually (Figure 7c,d), and the impact toughness increased as well, which is consistent with the test results of the impact test. The main reasons for the increase in the dimension of the dimples are the growth of the carbides and the decrease in the dislocation density during the tempering. At the same time, the dimples of unequal distribution and size were replaced by the equal-axis dimples [44].

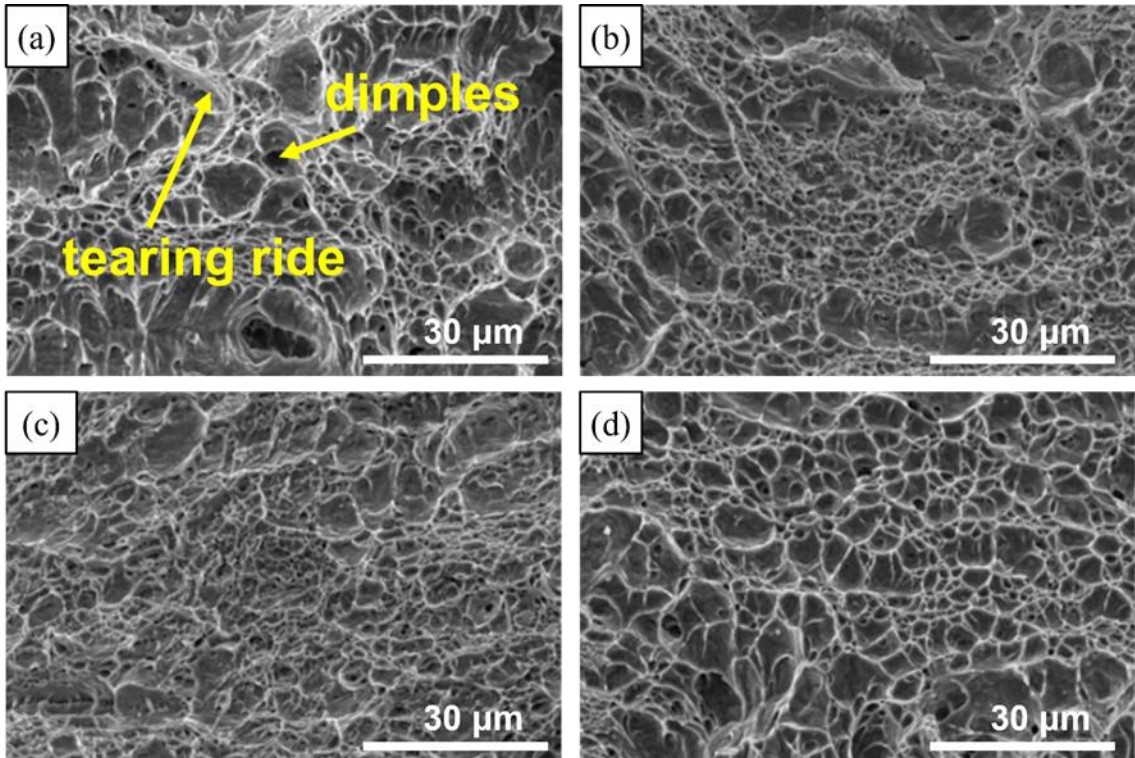

**Figure 7.** Cryogenic impact fracture morphologies of the quenched 15CrNi3MoV steel tempered at (**a**) 350 °C, (**b**) 450 °C, (**c**) 550 °C and (**d**) 650 °C.

The effect of the carbide on the impact toughness is complex. At a low tempering temperature, the carbides precipitated along the interface of the martensitic lath during the decomposition of the martensite, which became the path of crack propagation and reduced the fracture strength of the grain boundaries. However, the phenomenon of the brittle fracture caused by the precipitation of the carbides did not appear. With the increase of the tempering temperature, the acicular carbides began to change into granular carbides, which caused the pinning effect of the acicular carbides to weaken. With the accumulation and spheroidization of the precipitated carbides, the embrittlement characteristic was ameliorated and the toughness of the steel was improved [45]. Moreover, the stress concentration of the granular carbides was small, so the micro-crack was not easy to produce, which resulted in a good ductility of the steel.

The effect of alloying elements on impact toughness is more intricate than the carbides. The alloying elements, such as Cr and Ni in the steel, promoted the segregation of the impurity elements, and they were also segregated to the grain boundaries, which further reduced the strength of the fracture of the grain boundary, thereby increasing the tendency of brittleness [46]. The element of Mo interacted with the impurity elements to inhibit the segregation of the impurity elements to the grain boundaries, thereby reducing the tendency of brittleness. The influence of the alloying elements is closely related to their content [47].

In the 15CrNi3MoV steel, whether the alloying elements in the 15CrNi3MoV steel have a positive or negative effect on the comprehensive effect of toughness and plasticity also needs to be further explored. Since there was no brittle fracture, we tentatively believe that the comprehensive effect is positive.

## 4. Conclusions

1.  With the increase of the tempering temperature, the martensite gradually decomposed into a ferrite matrix and carbides, and the martensite and the ferrite matrix recovered and recrystallized, resulting in the martensitic lath boundaries gradually becoming blurred. The dislocation density between the laths gradually decreased and softened the matrix, which led to a decrease in strength.

2.  The dispersed carbides were preferentially precipitated at the original lath boundaries. Firstly, the precipitated carbides were acicular (about 130 nm in length), and then with the increase of the tempering temperature, the carbide gradually aggregated, grew and spheroidized. When the tempering temperature reached 650 °C, the carbides grew into short rodlike or granular particles (about 70 nm in length). The precipitates after the tempering at different temperatures were the $M_3C$ carbides, and their components are $Fe_3C$.

3.  With the increase of the tempering temperature, the yield strength, tensile strength and hardness all showed a decreasing trend. When tempered at 550 °C, the yielding to tensile ratio reached its maximum value (0.9253). The impact energy, the tensile strength and the yield strength decreased slowly in 350–550 °C and sharply in 550–650 °C, while the opposite was observed in the elongation and hardness. There was no phenomenon of brittle fractures after the tempering at different temperature, all of which were ductile fractures. Considering the rapid change in the mechanical properties in 550–650 °C and the practical engineering applications of the 15CrNi3MoV steel, the optimal tempering temperature is about 600 °C.

**Author Contributions:** Data curation, X.Y. and J.H.; Writing—original draft, X.Y., B.W. and B.X.; Writing—review & editing, Y.Q. and M.S. All authors have read and agreed to the published version of the manuscript.

**Funding:** This work is funded by the National Key Research and Development Program (No. 2018YFA0702900), the National Science and Technology Major Project of China (No. 2019ZX060040100), the Key Program of the Chinese Academy of Sciences (No. ZDRW-CN-2017-1), and the CAS Interdisciplinary Innovation Team.

**Institutional Review Board Statement:** Not applicable.

**Informed Consent Statement:** Not applicable.

**Data Availability Statement:** The data used to support the findings of this study are available from the corresponding author upon request.

**Conflicts of Interest:** The authors declare no conflict of interest.

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
