# Peer review of "Precipitation Behavior of Carbides and Its Effect on the Microstructure and Mechanical Properties of 15CrNi3MoV Steel"

_metals, doi:10.3390/met12101758_

Round 1
Reviewer 1 Report
The manuscript is describing the Precipitation behavior of carbides and its effect on the micro- 2 structure and mechanical properties of 15CrNi3MoV steel. the manuscript is well organized and very interesting but I have some requests that can further improve the quality of the manuscript. This does not under-estimate the manuscript value but will further improve the quality of the presented work.
1) TEM micrograph in Fig. 3 shows the precipitates evolution during tempering as stated these are carbides. however, further examination by diffraction pattern or EDX mapping can show us more information about the evolution of these precipitates during the tempering treatment.
2) Further work on work hardening related to data presented on Table 2 will be very interesting to the readers.
Author Response
Dear Reviewer,
I am very glad that you read my manuscript so carefully and gave such valuable opinions. Based on your opinions, I have made the following modifications:
- TEM micrograph in Fig. 3 shows the precipitates evolution during tempering as stated these are carbides. however, further examination by diffraction pattern or EDX mapping can show us more information about the evolution of these precipitates during the tempering treatment.
SEAD mapping has been added to Fig.3 to explain the evolution of these precipitates during the tempering treatment. And the XRD has shown that the carbides were Fe3C. Please see Fig.3 and Fig.4.
- Further work on work hardening related to data presented on Table 2 will be very interesting to the readers.
Further research will be carried out on the specific influence of yielding to tensile ratio and its optimization. However, it is not explained too much because the current date is not enough and the research on the yielding to tensile ratio is not systematic. In addition, a literature on the yielding to tensile ratio has been added to increase the readability of the article. Please see line 238.
Reviewer 2 Report
Comments to Authors
This article deals with presented and studied precipitation behavior of carbides and its effect on the microstructure and mechanical properties of 15CrNi3MoV steel, which is an interesting topic.
This work is well-written and of high scientific merit. The suitability of the research procedures was employed, as well as a thorough understanding of research methodology. The research work was properly planned and carried out. For the kind of literature discussed in this article, in-depth knowledge of the area is required. The paper's results and interpretation should be enhanced.
After reading the manuscript, I can conclude that this paper may be accepted with a Minor revision.
However, few modifications are needed to enhance the quality of the manuscript further.
- Abstract of the article is not clear and concise. The abstract part needs to include mathematical findings to be more informative.
- The title of the manuscript is to be corrected in revision.
- In the introduction, add recent literature published after 2018.
- Results and discussion must be supported by standard literature.
- Figures 1, 2, and 3, provide good resolution. It is hard to see and investigate. High-quality Figures to be provided for better readability with proper legend and labels.
- The conclusion is needed to write more precisely with the application of these existing methodology.
Author Response
Dear Reviewer,
I am very glad that you read my manuscript so carefully and gave such valuable opinions. Based on your opinions, I have made the following modifications:
- Abstract of the article is not clear and concise. The abstract part needs to include mathematical findings to be more informative.
Abstract has been modified.
2. The title of the manuscript is to be corrected in revision.
The title has been corrected.
3. In the introduction, add recent literature published after 2018.
The recent literature have been added. Please see line 67-76.
4. Results and discussion must be supported by standard literature.
The relevant literature have been supported. If you have any good literature to referent, please tell me kindly.
5. Figures 1, 2, and 3, provide good resolution. It is hard to see and investigate. High-quality Figures to be provided for better readability with proper legend and labels.
Figures have been modifided.
6. The conclusion is needed to write more precisely with the application of these existing methodology.
The conclusion has been revised. Please check it and give me more good ideas.
Best regaard
Reviewer 3 Report
See attachment

Author Response
Dear Reviewer,
I am very glad that you read my manuscript so carefully and gave such valuable opinions. Based on your opinions, I have made the following modifications:
row 17 trend.The trend. The
whole text 650°C and so on 650 °C (by International System of Units)
row 33 thermos-mechanical thermo-mechanical
row 54 873K 873 K (by SI units)
row 121 quenched quenching
row 225 low alloy steel low alloyed steel
row 288 steel, Whether steel, whether
All of the mistakes have been corrected. Please see the revised manuscript.
Reviewer 4 Report
Dear authors, you can see my comments in the attached file.
Best regards!

Author Response
Dear Reviewer,
I am very glad that you read my manuscript so carefully and gave such valuable opinions. Based on your opinions, I have made the following modifications:
1) In line 53 the authors write the following: “[22] studied the strengthening mechanism of carbide MX”. I believe that it would be a good idea to add the meaning of such“carbide MX” as in the paper [22] is. If that information is added the potential readers can know what it is without consulting [22]. I have such information, but not everyone can access it!
It’s a good idea to add the meaning of “carbide MX”. The meaning (Carbonitrides of V and Nb) has been added.Please see line 53.
2) In line 110 the authors write about a “a strain rate of 3 mm·s-1 was employed”. With relation to this I think that the authors want to write the following: “a displacement rate of 3 mm·s-1 was employed”. I believe, sorry if I am wrong, that the tension test was performed under displacement control of the moving crosshead of the testing machine (I believe that due to the units of mm·s-1 indicated by the authors). It is also true that the test can be performed under strain control; in this case the control of the tensile test is ruled by the extensometer (more difficult to perform without risk for the testing equipment), in this last case the units can not be expressed as mm·s-1, should be referred as s-1.
Thank you for your reminding. If it hadn't been for your reminding, I would have overlooked this crucial mistake. In my experiment, a displacement rate of 3 3mm/min was employed. Please see line 120.
3) With relation to Fig. 1: the authors write that the prior austenite grains are visible in Fig. 1(a). In my opinion the PAGs are possible to see in Fig. 1(b) but not (very difficult) in Fig. 1(a).
I am sorry that the PAGs in Fig.1(a) ware not clear. The PAGs have been marked. If that are not PAGs in your opinions, please tell me.
4) Can the authors to include a brief description of the term “self temper” (line 127)?
Self temper is a tempering phenomenon that occurs when the specimen is not completely cooled during the quenching process due to dimensional effects. Please see line 137.
5) With relation to Fig. 2: to be honest I believe that the four microstructures showed are not the best choice to see the details. Let me explain this; it is very difficult to see in the figures that it is inside the corresponding text. The microstructure of Fig. 2(b) can be confused with Fig. 2(c) and vice versa. Figs. 2(c) and (d) are not cited in the text. In my opinion the photos should be made at higher magnification, and more clear, to see the differences between them.
The figures have been optimized. The lath martensite boundaries in Figs. 2(c) were a little more ambiguous than that in Figs. 2(b). Although this difference is not particularly obvious, it is the real result in the experiment. Figs. 2(c) and (d) are not cited in the text in line 162.
6) In line 165 and with relation to Fig. 3(a): the authors write that “the acicular
precipitates have a width about 12 nm and length about 150 nm,...”. With relation to the length of the acicular precipitates inside the Fig. 3(a) I believe that the average length of such acicular precipitates is clearly lower than 150 nm. Sorry if I am wrong.
According to your suggestion, I used MATLAB to make size statistics again, and found that the average length was not 150nm, which is the same as you said. Maybe the statistics I made last time were not complete, and the large size was selected for the statistics. It’s my mistake. According to the latest statistics, the average length is 120nm. Please see line 176 and 182.
7) With relation to the point 3.3. The effect of microstructure on mechanical properties. I think that this point is very interesting and important. It is a pity that the authors do not treat the toughness of the steel (the area under the stress-strain curve) for the different cases studied. The toughness of a steel like the present one is a mechanical property very important to consider.
This article focuses on strength and toughness. Although experiments related to elongation had been done, they were only used as a reference and do not focus on stress-strain curves related to plasticity.
8) Between the lines 241-244 it is possible to read the following: “At tempering
temperature above 550°C, the impact energy increased dramatically, and thus, a
considerably increase of energy absorbed with increased tempering temperature was observed. This finding is consistent with the result of M.F. de Souza et al. [39]”. Here we have two sentences. The content of the first sentence is true, and it is well known from several decades ago (it is a general trend inside the Metallic Materials Science). Due to that I believe that the presence second sentence it is not necessary: nowadays, it is impossible to find a procedure to increase both values: the impact energy and the hardness. If one of then increase the other decrease (and vice versa). Sorry if I am wrong, really.
You are right, that’s a general trend inside the Metallic Materials Science. The relevant sentence has been deleted.
9) Between the lines 247-249 there are two sentences, I believe that the authors can erase the word “probably” in the second sentence. The four images in Fig. 3 are very clear (the precipitation and coarsening of the precipitates are evident!).
The word ”probably” has been deleted.
10) Line 244: inside Fracture Mechanics world it is more adequate, more common, to use the term microvoids instead micropores. It is just an advice.
Your profound learning and academic rigor are admirable. The term micropores have been replaced by microvoids. Please see line 271.
11) Between lines 262-263: the term toughness should be written as impact toughness (to avoid possible confusion with others “toughness”: that from tensile test -mainly- or that from fracture toughness).
Thank you for your remind. Your suggestion can avoid many unnecessary misunderstandings. The term toughness has been written as impact toughness. Please see line 280.
12) Between lines 263-265 the authors write the following “The main reason of the increase in the dimension of dimples is the growth of carbides during tempering”. The sentence is right, but we can no forget that the plastic behaviour of the metallic alloy, the matrix, was increased due to the tempering’s. So, the main reason of the increase in the size of the dimples (or voids) are two: the growth of the carbides due to the tempering process, and the increase of the ability to support plastic strains of the bulk material due to the mentioned tempering process (thanks to the great decrease in the dislocations density, mainly, and more things).
Thank you for your reminding. It is true that the plastic behavior of the metallic alloy should not be ignored in fracture behavior. In the previous text, it was mentioned that the matrix softened and the dislocation density decreased, but I missed this point here. It has now been corrected, please see line 281-282.
13) Finally I want to write that the term sorbite is obsolete from several decades ago (from the 70’s of the last century if I remember well). The term sorbite it is obsolete, and this is a matter of fact (sorry, but this is true!). I perfectly know that the term sorbite is used today by several authors (like in books of several decades ago), but this does not imply that they, authors and books, are right. The term sorbite was coined by Sorby in 1883 (published later in 1886) because he thought he had discovered a new steel’s structure. A few decades later some researchers, with new metallographic techniques, demonstrate the importance of a good and adequate specimen preparation to observe its microstructure correctly. Numerous new and well-known scientific books around the world about Materials Science write that sorbite is an obsolete term and says that:
“modern metallurgist describes these microstructures as very fine pearlite or as
tempered martensite”. As an example, it’s possible to find inside the ASM-HandBook collection, volume 9 named Metallography & Microstructures (a very well-known reference book), the following: At the end of the 19th century, very fine pearlite unresolved in the light microscopes was referred to as “sorbite” in honor of Sorby. However, because it is not a new constituent, the term “sorbite” did not survive. To conclude: the term sorbite is obsolete and has no sense from several decades ago because sorbite is equal to tempered martensite (if it’s obtained by tempering after hardening), and sorbite is equal to very fine pearlite (if it’s obtained by regulating the cooling of the steel), it’s so fine that it is very difficult to note that under traditional microscopes (this was the original mistake from Sorby). Let’s end the use of this obsolete term together! I perfectly know that sorbite is used today by several authors but... are they doing the right thing using such outdated term? We belong to the scientific community of Materials Science, and we must be rigorous. As far as I know the term sorbite is obsolete, maybe I am wrong, maybe in the last years the term sorbite has been resurrected by the right or relevant authority (I’m afraid not, sorry if I’m wrong!). I perfectly know that the following sentence can be considered pretentious, the sentence in question is: today the term sorbite is used in a wrong way due to a snowball effect, that is, I use it because they use it (so the snowball is growing
up!). To end: if the term sorbite is obsolete, the term tempered sorbite is also unknown and obsolete!
I'm sorry that my negligence and lack of rigor may have confused the reader with the sorbite. After literature reading and further study, it is found that the sorbite and tempered sorbite in this paper can be classified as martensite and tempered martensite. I apologize again for my ignorance and lack of rigor.
Kind regards.
Reviewer 5 Report
This paper describes a study of the effects of tempering temperature on the microstructure and mechanical properties of a high-strength low-alloy steel. The authors heat treated the steel specimens at a range of temperatures from 350 – 650 °C and examined the microstructures, as well as conducting tensile, hardness and fracture toughness tests. They found that increasing the tempering temperature reduced the hardness, yield stress and tensile strength, while increasing the toughness and elongation. These findings are in agreement with previous studies reported by other workers on steels of similar composition, so the novelty of the present study is slight. However, they have used TEM to show what they state to be carbides (not reported in other studies), which appear to show the effect of temperature on the carbide morphology.
There are a number of changes that must be made before the paper can be accepted. They are listed below.
1. Section 2.3 – please state how the strain was measured in the tensile tests.
2. In Section 3.2, the authors state that they observed “fine carbide precipitation in the interlath regions after tempering”. Where are these carbides? They are not obvious from the images in Figure 2.
3. In Figure 3, the authors state that the precipitates in the TEM images are carbides. What techniques have they used to confirm this, apart from the images shown? Although the XRD patterns in Figure 4 indicate the presence of carbides, XRD cannot identify the precipitates in Figure 3 as carbides. A higher resolution technique, such as EDS / WDS is needed.
4. Figure 4 – please give the X-ray powder diffraction file number for the carbide. Also, it would be better if the carbide could be referred to as “Fe3C” rather than the generic “M3C”.
5. Figure 5 – please include error bars for the data points in this figure.
6. Section 3.3, line 209: the yield stress and tensile strength both decrease monotonically with increasing temperature.
7. Section 3.3, line 225: please provide a reference for the statement that the yield stress to tensile strength ratio is between 0.65 and 0.75. At what temperature is this true?
8. The heading on line 250 should be section 3.4.
9. The authors report that the hardness, yield stress and tensile strength decrease with increasing tempering temperature, while the elongation and toughness increase. This has been reported in other studies, for example Salemi and Abdollah-Zadeh (Materials Characterization, 59 (2008), 484-487) and Zhu et al., IOP Conf. Ser.: Mater. Sci. Eng., 493 (2019), 012141). The authors should reference these studies in their discussion of the results.
In addition, there are a number of typographical or grammatical changes that are needed to the manuscript: they are listed below.
· Introduction, line 33: please change to “…thermo-mechanical…”
· Introduction, line 35: please change to “Metallurgists…”
· Section 3.1, line 121: please change to “…quenching from 890 °C…”
· Section 3.3, line 219: please change to “…led to the increase…”
· Section 3.3, line 221: please change to “…led to the ratio decreasing.”
· Section 3.3, line 235: please change to “…increase in elongation”.
· Section 3.3, line 240: please change to “The values of mean impact energy…”
· Section 3.3, line 242: please change to “…considerable increase of energy…”
· Section 3.3, line 247: please change to “Increasing the tempering temperature…”
· Section 3.3, line 248: please change to “…led to a significant decrease…”
· Section 3.4, line 256: please change to “…continuous or discontinuous carbides…”
· Section 3.4, line 264: please change to “…reason for the increase…”
Author Response
Dear Reviewer,
I am very glad that you read my manuscript so carefully and gave such valuable opinions. Based on your opinions, I have made the following modifications:
1. Section 2.3 – please state how the strain was measured in the tensile tests.
Elongation is measured by mechanical extensometer.Please see line 122.
2. In Section 3.2, the authors state that they observed “fine carbide precipitation in the interlath regions after tempering”. Where are these carbides? They are not obvious from the images in Figure 2.
The signs have been marked. Please see Fig.2 .
3. In Figure 3, the authors state that the precipitates in the TEM images are carbides. What techniques have they used to confirm this, apart from the images shown? Although the XRD patterns in Figure 4 indicate the presence of carbides, XRD cannot identify the precipitates in Figure 3 as carbides. A higher resolution technique, such as EDS / WDS is needed.
SEAD mapping has been added to Fig.3 to explain the evolution of these precipitates during the tempering treatment. Please see Fig.3.
4. Figure 4 – please give the X-ray powder diffraction file number for the carbide. Also, it would be better if the carbide could be referred to as “Fe3C” rather than the generic “M3C”.
M3C has been changed to Fe3C. And the number of alloy cementite is ICSD#64689.
5. Figure 5 – please include error bars for the data points in this figure.
I am sorry that just one set of experiments had been done to look at trends rather than specific numbers. I am sorry again that I have no enough samples to do more experiments to include error bars.
6. Section 3.3, line 209: the yield strength and tensile strength both decrease monotonically with increasing temperature.
Thank you for your reminding. The mistake has been corrected. Please see line 221.
7. Section 3.3, line 225: please provide a reference for the statement that the yield stress to tensile strength ratio is between 0.65 and 0.75. At what temperature is this true?
Sorry, it's my error to forget the reference. Now the reference has been provided, and the temperature is service operating temperature, which is room temperature in most cases.
8. The heading on line 250 should be section 3.4.
Sorry, it’s my mistake. The heading has been revised. Please see line 267.
9. The authors report that the hardness, yield stress and tensile strength decrease with increasing tempering temperature, while the elongation and toughness increase. This has been reported in other studies, for example Salemi and Abdollah-Zadeh (Materials Characterization, 59 (2008), 484-487) and Zhu et al., IOP Conf. Ser.: Mater. Sci. Eng., 493 (2019), 012141). The authors should reference these studies in their discussion of the results.
The literature you provided has inspired me a lot. At your suggestion, I have referenced more studies in their discussion of the results. Please see line 66-76.
10.
In addition, there are a number of typographical or grammatical changes that are needed to the manuscript: they are listed below.
- Introduction, line 33: please change to “…thermo-mechanical…”
- Introduction, line 35: please change to “Metallurgists…”
- Section 3.1, line 121: please change to “…quenching from 890 °C…”
- Section 3.3, line 219: please change to “…led to the increase…”
- Section 3.3, line 221: please change to “…led to the ratio decreasing.”
- Section 3.3, line 235: please change to “…increase in elongation”.
- Section 3.3, line 240: please change to “The values of mean impact energy…”
- Section 3.3, line 242: please change to “…considerable increase of energy…”
- Section 3.3, line 247: please change to “Increasing the tempering temperature…”
- Section 3.3, line 248: please change to “…led to a significant decrease…”
- Section 3.4, line 256: please change to “…continuous or discontinuous carbides…”
- Section 3.4, line 264: please change to “…reason for the increase…”
Thank you very much for reading it so carefully, these typographical or grammatical problems above have been changed.
If you have any other advise, please tell as soon as possible. Your suggestions are very valuable for me.
Best regards